# The cell surface protein MUL_3720 confers binding of the skin pathogen *Mycobacterium ulcerans* to sulfated glycans and keratin

Christopher J. Day[1], Katharina Röltgen[2,3], Gerd Pluschke[2,3]*, Michael P. Jennings[1]*

**1** Institute for Glycomics, Griffith University, Gold Coast, Queensland, Australia, **2** Swiss Tropical and Public Health Institute, Basel, Switzerland, **3** University of Basel, Basel, Switzerland

* Gerd.Pluschke@swisstph.ch (GP); m.jennings@griffith.edu.au (MPJ)

**Data Availability Statement:** All relevant data are within the manuscript and its Supporting Information files.

## Abstract

*Mycobacterium ulcerans* is the causative agent of the chronic, necrotizing skin disease Buruli ulcer. Modes of transmission and molecular mechanisms involved in the establishment of *M. ulcerans* infections are poorly understood. Interactions with host glycans are often crucial in bacterial pathogenesis and the 22 kDa *M. ulcerans* protein MUL_3720 has a putative role in host cell attachment. It has a predicted *N*-terminal lectin domain and a *C*-terminal peptidoglycan-binding domain and is highly expressed on the surface of the bacilli. Here we report the glycan-binding repertoire of whole, fixed *M. ulcerans* bacteria and of purified, recombinant MUL_3720. On an array comprising 368 diverse biologically relevant glycan structures, *M. ulcerans* cells showed binding to 64 glycan structures, representing several distinct classes of glycans, including sulfated structures. MUL_3720 bound only to glycans containing sulfated galactose and GalNAc, such as glycans known to be associated with keratins isolated from human skin. Surface plasmon resonance studies demonstrated that both whole, fixed *M. ulcerans* cells and MUL_3720 show high affinity interactions with both glycans and human skin keratin extracts. This MUL_3720-mediated interaction with glycans associated with human skin keratin may contribute to the pathobiology of Buruli ulcer.

## Author summary

*Mycobacterium ulcerans* causes a skin-based disease known as Buruli ulcer. How the bacteria are transmitted and what mechanisms they use to establish the infection of the skin is poorly understood. The only well characterized bacterial factor in Buruli ulcer pathogenesis is mycolactone, a toxin produced by the bacteria. Mycolactone causes apoptosis in human cells, leading to destruction of the skin around extracellular clusters of the mycobacteria. Human cells, like cells of all orders of life, are coated in complex sugar structures and these glycans are one of the major targets of bacteria and viruses for the interaction with host cells. Here we describe the glycan binding of whole *Mycobacterium ulcerans* cells and a mycobacterial protein, MUL_3720, thought to be involved in glycan binding. We show that both the bacterial cells and MUL_3720 bind to glycans known to be associated with human skin keratin and to skin keratin extracts. This binding of keratin extracts

**Funding:** This work was supported by a National Health and Medical Research Council (NHMRC; https://www.nhmrc.gov.au/) Program Grant (1071659 to MPJ), NHMRC Principal research fellowship (APP1138466 to MPJ) and a grant of the Medicor Foundation (to GP; https://medicor.li/en/). The funders had no role in study design, data collection and analysis, decision to publish, or preparation of the manuscript.

**Competing interests:** The authors have declared that no competing interests exist.

may explain initial bacterial attachment and clustering of the bacteria in the skin, ultimately leading to tissue destruction and ulceration caused by a cloud of secreted mycolactone at the site of infection.

## Introduction

Buruli ulcer (BU) is a chronic, necrotizing skin disease, caused by *Mycobacterium ulcerans* [1]. It affects populations living in contact with stagnant or slow flowing water bodies primarily in West and Central Africa, but has also been reported from Asia, the Americas, Papua New Guinea and Australia [2]. The interaction of *M. ulcerans* with the human host is not fully understood. The site of inoculation of the bacteria into susceptible layers of the skin is thought to be the site where an infection is established. Presence of only one lesion in the majority of BU patients speaks for a very low rate of contiguous spread. Thermo-sensitivity of the pathogen may in part explain why infections are typically confined to the cooler surface of the body rather than internal organs, which are not infected [3]. Production of the macrolide toxin mycolactone by *M. ulcerans* [4] causes apoptosis in mammalian cells [5] and leads to extensive tissue necrosis. In advanced BU lesions, extracellular clusters [6] of the pathogen are residing in completely necrotic areas, primarily localized in deeper layers of subcutaneous fat tissue [7]. A protective cloud of mycolactone appears to prevent infiltrating leukocytes to reach the bacteria [8]. Clustering of the bacteria and skin location thus appear key elements in the long-term persistence of *M. ulcerans* in the chronically infected immunocompetent host.

Many studies indicate a key role for host carbohydrates as targets for bacterial adhesins [9]. Here we conducted a glycomic analysis to define the glycan-binding repertoire of whole, fixed *M. ulcerans* cells. In a second step, we compared this repertoire with the glycan-binding activity of the candidate adhesin MUL_3720. 2D gel electrophoretic analysis of an *M. ulcerans* whole protein lysate has shown that MUL_3720 is one of the most highly expressed proteins of the pathogen, a feature that is currently being exploited in the development of a diagnostic antigen capture assay [10]. The function of MUL_3720 is suggested by its two-domain structure, with a conserved bulb-type lectin domain and a Lysine Motif (LysM) domain, which is predicted to be involved in binding to peptidoglycan [11]. Studies with a mycobacterial-specific two-hybrid system furthermore indicated that MUL_3720 interacts with a range of cell wall associated proteins [10]. Immunofluorescence staining of *M. ulcerans* bacilli demonstrated a cell wall localization of MUL_3720 [10,12] and all these features together suggested that MUL_3720 plays a role in the binding of *M. ulcerans* cells to glycosylated substrates. Here we show that MUL_3720 is binding to sulfated galactose and GalNAc structures, which are elements of sulfated glycosaminoglycans (GAGs), such as chondroitin sulfate, dermatan sulfate and heparan sulfate.

## Materials and methods

### Growth and maintenance of *M. ulcerans*

The *M. ulcerans* strain S1013, recently isolated from the lesion of a BU patient from Cameroon [13] was grown in BacT/Alert culture bottles supplemented with enrichment medium according to the manufacturer's protocol (bioMérieux). Cells were fixed with 4% formalin prior to array and SPR analysis.

## Cloning, expression, and purification of MUL_3720

Full length MUL_3720 (aa 1–207) was cloned into pET28a, as previously outlined [14]. The recombinant protein was expressed and purified as previously described [10,14], with the purity on par with that shown in Bolz *et al*. 2016 [14].

## Evolutionary analysis of the MUL_3720 protein

The MUL_3720 protein sequence was analysed using the BLAST tool (NCBI) against the non-redundant protein sequence database and mycobacterial sequences with an e-value of > 1e-30 were downloaded for further analysis against the MUL_3720 sequence. The evolutionary history of the retrieved sequences was inferred using the Maximum Likelihood method and Whelan And Goldman + Freq. model [15]. Initial tree(s) for the heuristic search were obtained automatically by applying Neighbour-Join and BioNJ algorithms to a matrix of pairwise distances estimated using a JTT model, and then selecting the topology with superior log likelihood value. A discrete Gamma distribution was used to model evolutionary rate differences among sites (5 categories (+*G*, parameter = 0.5861)) of an 88 amino acid sequence region of the proteins. Evolutionary analyses were conducted in MEGA X [16,17] and shown in S1 Fig.

## Glycan array analysis

Glycan arrays were printed onto SuperEpoxy 3 activated substrates as previously described [18]. The glycan array analysis of the whole, fixed *M. ulcerans* cells was performed using a modification of the method used in Day et al 2009 and Mubaiwa et al 2017 [19,20]. Briefly, approximately $10^6$ Bodipy 595/625 nm labelled bacteria in 500 μL of array PBS (1 x PBS containing 1 mM $CaCl_2$ and 1 mM $MgCl_2$) were applied to the glycan array in a 65 μL gene frame without a cover. The bacteria were incubated on the array at room temperature in the dark for 30 minutes and washed three times for five minutes, once in array PBS and twice in 1x PBS without Mg or Ca ions. Glycan arrays with the MUL_3720 protein were performed as previously described [21] using 2 μg of protein per array and 20 minutes of incubation time prior to washing. Both protein and cell arrays were scanned and analysed using the Innoscan 1100AL scanner and the MAPIX analysis package. Final statistical analysis of the combined datasets were performed in Microsoft Excel using Student's T test. The lis of glycans is present in S1 Data and the MIRAGE compliance information is included in S1 Table.

## Surface plasmon resonance analysis

Surface plasmon resonance (SPR) analyses were performed as previously described on a Biacore T200 using a Series S C1 chip for whole bacteria analysis [20,22] and a CM5 chip for protein analysis [21] with the following modifications. Proteins were immobilized onto a CM5 chip at pH 4.5, flow rate of 5 μL/min for 600 seconds with an ethanolamine blank flow cell as a control. The C1 chip was prepared as per manufacturer's instructions and 100 μL of bacteria at approximately $10^6$ cells per mL at pH 4.5 were immobilized using amine chemistry at a flow rate of 5 μL per minute for 720 seconds. Glycans (Fucα1-2Gal, Fucα1-2Galβ1-4(Fucα1–3) GlcNAc, Neu5Acα2-6Galβ1-4GlcNAc, Fucα1-2(3-O-Su)Galβ1-3(Fucα1–4)GlcNAc; Elicityl, mannose glycans; Dextra Laboratories), GAGs (GAGs; Dextra Laboratories) and keratin isolated from human epidermis (mixed keratins with average molecular weight of 64.1 kDa; Sigma-Aldrich, Cat# K0253) were tested between 2 nM and 20 μM. All data was double reference subtracted. Analysis was performed using the Biacore T200 Evaluation software package. Competition assays were performed using the sulphated chondroitin disaccharide ΔUAα1-3GalNAc-6S and a keratin extract, which were flowed separately or together at a maximum of

10 times the calculated $K_D$ (1μM for ΔUAα1-3GalNAc-6S and 50nM for keratin extract) to ensure saturation in direct competition using a modification of a previously described method [20]. Maximum responses for each interaction were used as final point data for the interaction and compared to the calculated average and addition to determine the presence of a shared or separate binding site.

### N-linked glycan endoglycosidase PNGase F treatment of keratin from human epithelial cells

Keratin isolated from human epidermis (10 μg mixed keratins with average molecular weight of 64.1 kDa; Sigma-Aldrich, Cat# K0253) was treated with PNGase F (New England Biolabs) under denaturing conditions as described in the manufacturers protocol. The same quantity of keratin was incubated in the same buffers for the same time without PNGase F as a control. Half of each reaction, plus/minus PNGase, were run on 4–12% PAGE gels in duplicate and the gels were cut in half and stained using different methods. One gel half was stained with the ProteoSilver Silver Stain Kit (Sigma) following manufacturers protocol for the staining of proteins. The other portion of the gel was stained using the ProteoSilver Silver Stain Kit (Sigma) following the method for the staining of sugars as previously described [23]. A DNA ladder was run as a control for the sugar staining and a NEB 11-250kDa prestained blue protein ladder was run as a control for the protein size and staining.

## Results

### *M. ulcerans* cells and MUL_3720 bind to a subset of sulfated glycans

MUL_3720 is annotated as a possible lectin, based on the presence of a B-lectin domain that has a mannose binding consensus sequence [10], to our knowledge the only annotated lectin in *M. ulcerans*. Due to this hypothesised lectin activity, both MUL_3720 and whole fixed *M. ulcerans* cells were analysed for glycan binding activity using a glycan arrays. This analysis revealed that the *M. ulcerans* cells bind to 64 of the 368 glycans printed on the array (Table 1 and S1 Data). We reasoned that MUL_3720 may be responsible for part of these interactions and found that it is binding to seven glycans on the array (Table 1, S1 Data). The *M. ulcerans* cells bound to a range of distinct glycans including mannose structures, Lewis and ABO blood group antigens, sialylated glycans and sulfated glycans including both sulphated GAGs and sulfated galactose and GalNAc structures (Table 1). MUL_3720 only recognized sulfated galactose and GalNAc structures including those typical of the glycosylation present on keratins and GAGs (Table 1) [24–26].

### *M. ulcerans* cells and MUL_3720 bind to specific glycans with high affinity

To validate the glycan array results using a different technique and to determine the dissociation equilibrium constant ($K_D$) of the interactions, SPR analyses were performed between free oligosaccharides and *M. ulcerans* cells or purified recombinant MUL_3720 that were bound to sensor chips (Table 2). Eight glycan structures representative for each of the distinct glycan groups that either bound only to *M. ulcerans* cells or to both *M. ulcerans* cells and purified MUL_3720 were chosen. These structural groups are representatives of blood group antigens (Fucα1-2Gal), Lewis antigens (Fucα1-2Galβ1-4(Fucα1–3)GlcNAc), sulfated glycans (GAGs including chondroitin 6S-complex polymer, chondroitin 62 disaccharide ΔUAα 1-3GalNAc-6S and, heparin and Fucα1-2(3-O-Su)Galβ1-3(Fucα1–4)GlcNAc), mannose structures (Manα1-3Man, Manα1-6(Manα1–3)Manα1-6(Manα1–3)Man) and sialylated glycans (Neu5Acα2-6Galβ1-4GlcNAc). This set of structures provides a broad coverage of the glycan

**Table 1. Glycans bound by *M. ulcerans* cells and MUL_3720 protein in glycan array analysis.** Red labelling indicates binding as determined by interaction above background in three replicate array experiments. A positive fluorescent value is defined as any value above the average background fluorescence of 20 negative control spots + 3 standard deviations. Data are presented as fold-above this background value, which were 2706±372 RFU for *M. ulcerans* cells and 1829±140 RFU for the MUL_3720 studies. Only glycans that showed binding are shown; see S1 Data for full array results and list of all the glycans present on the array.

| Array ID | Structure | *M. ulcerans* | MUL_3720 |
|---|---|---|---|
| | **Terminal Galactose Structures** | | |
| 145 | Galβ1-3(6-O-Su)GlcNAcβ-sp3 | 1.92±0.64 | 0.432099 |
| 146 | Galβ1-4(6-O-Su)Glcβ-sp2 | 1.59±0.18 | 0.382383 |
| 147 | Galβ1-4(6-O-Su)GlcNAcβ-sp3 | 1.94±0.34 | 0.441459 |
| 151 | 6-O-Su-Galβ1-3GalNAcα-sp3 | 1.87±0.40 | 2.44±0.88 |
| 152 | 3-O-Su-Galβ1-4Glcβ-sp2 | 1.91±0.98 | 0.432046 |
| 159 | 4-O-Su-Galβ1-4GlcNAcβ-sp3 | 1.73±0.26 | 0.234259 |
| 161 | 6-O-Su-Galβ1-3GlcNAcβ-sp3 | 1.64±0.37 | 2.49±0.26 |
| 178 | 6-O-Su-Galβ1-4(6-O-Su)Glcβ-sp2 | 1.17±0.26 | 0.442476 |
| 179 | 6-O-Su-Galβ1-3(6-O-Su)GlcNAcβ-sp2 | 1.60±0.41 | 1.48±0.33 |
| 180 | 6-O-Su-Galβ1-4(6-O-Su)GlcNAcβ-sp2 | 1.83±0.51 | 0.456808 |
| 182 | 3,6-O-Su$_2$-Galβ1-4GlcNAcβ-sp2 | 1.99±0.62 | 3.29±1.77 |
| 189 | 3,6-O-Su$_2$-Galβ1-4(6-O-Su)GlcNAcβ-sp2 | 1.90±0.45 | 2.46±0.56 |
| 201 | 3,4-O-Su$_2$-Galβ1-4GlcNAcβ-sp3 | 1.92±0.92 | 1.52±0.64 |
| 203 | Galβ1-4(6-O-Su)GlcNAcβ-sp2 | 1.92±0.79 | 0.457048 |
| 18C | Galβ1-3GalNAcβ1-3Gal | 1.77±0.94 | 0.488285 |
| | **Fucosylated glycans** | | |
| 215 | Fucα1-2Galβ1-3GlcNAcβ-sp3 | 1.86±0.26 | 0.361827 |
| 219 | Fucα1-2Galβ1-4Glcβ-sp4 | 1.88±0.93 | 0.42071 |
| 542 | Le$^c$Le$^x$1-6'(Le$^c$1-3')Lac-sp4 | 2.77±0.76 | 0.045969 |
| 7A | Fucα1-2Galβ1-3GlcNAcβ1-3Galβ1-4Glc | 1.63±0.92 | 0.325579 |
| 7E | Galβ1-3(Fucα1−4)GlcNAcβ1-3Galβ1-4(Fucα1−3)Glc | 1.82±0.72 | 0.459277 |
| 7G | Fucα1-2Galβ1-4Glc | 1.69±0.59 | 0.444367 |
| 7L | Fucα1-2Galβ1-4(Fucα1−3)Glc | 1.83±0.80 | 0.471871 |
| 7P | Fucα1-2Galβ1-3(Fucα1−4)GlcNAc | 1.66±0.26 | 0.473629 |
| | **Terminal *N*-Acetylgalactosamine** | | |
| 4 | GalNAcα-sp0 | 1.48±0.63 | 0.392032 |
| 193 | 3-O-Su-GalNAcβ1-4GlcNAcβ-sp3 | 1.52±0.95 | 0.261001 |
| 195 | 6-O-Su-GalNAcβ1-4-(3-O-Su)GlcNAcβ-sp3 | 1.93±0.86 | 0.409689 |
| 199 | 4,6-O-Su$_2$-GalNAcβ1-4-(3-O-Ac)GlcNAcβ-sp3 | 1.54±0.79 | 2.45±0.87 |
| 2C | Gal*N*Acβ1-3Gal | 1.76±0.40 | 0.414296 |
| 2F | GalNAcα1-3Galβ1-4Glc | 1.89±0.34 | 0.557756 |
| | **Mannose** | | |
| 19 | ManNAcβ-sp4 | 1.72±0.45 | 0.357543 |
| 5D | Manα1-3Man | 1.48±0.34 | 0.022 |
| 5E | Manα1-4Man | 1.56±0.81 | 0.213395 |
| 119 | Manα1-2Manβ-sp4 | 1.49±0.14 | 0.481129 |
| 122 | Manα1-6Manβ-sp4 | 1.49±0.28 | -0.02186 |
| 123 | Manβ1-4GlcNAcβ-sp4 | 1.69±0.54 | 0.47481 |
| | ***N*-Acetylglucosamine** | | |
| 118 | GlcNAcβ1-6GalNAcα-sp3 | 1.67±0.81 | 0.107263 |
| 149 | GlcNAcβ1-4(6-O-Su)GlcNAcβ-sp2 | 1.68±0.29 | -0.0567 |
| 493 | (GlcNAcβ1−4)$_5$β-sp4 | 1.47±0.32 | 0.24036 |
| 4C | Glc*N*Acβ1-4Glc*N*Acβ1-4Glc*N*Acβ1-4Glc*N*Ac | 1.44±0.59 | 0.137314 |
| 4E | Glc*N*Acβ1-4Mur*N*Ac | 1.48±0.73 | 0.04918 |

*(Continued)*

**Table 1.** (Continued)

| Array ID | Structure | *M. ulcerans* | MUL_3720 |
|---|---|---|---|
| 18G | 6-O-Su-GlcNAc | 1.30±0.675 | 0.441581 |
| | **N-Acetylneuraminic acid** | | |
| 18K | 9-NAc-Neu5Ac | 1.99±0.46 | 0.494373 |
| 18O | Neu5Gc | 1.60±0.59 | 0.464909 |
| 169 | Neu5Acα2-3Galβ-sp3 | 1.70±0.31 | 0.327169 |
| 170 | Neu5Acα2-6Galβ-sp3 | 1.77±0.18 | 0.516095 |
| 171 | Neu5Acα2-3GalNAcα-sp3 | 1.51±0.70 | 0.41582 |
| 172 | Neu5Acα2-6GalNAcα-sp3 | 1.68±0.34 | 0.470861 |
| 174 | Neu5Gcα2-6GalNAcα-sp3 | 1.77±0.24 | 0.469814 |
| 421 | GalNAcβ1-4(Neu5Acα2–3)Galβ1-4Glcβ-sp2 | 1.83±0.41 | 0.469452 |
| 533 | GalNAcβ1-4(Neu5Acα2-8Neu5Acα2-8Neu5Acα2–3)Galβ1-4Glc-sp2 | 1.62±0.77 | 0.468232 |
| | **Glucose** | | |
| 9 | Glcβ-sp3 | 1.63±0.71 | 0.457861 |
| 164 | GlcAβ1-3GlcNAcβ-sp3 | 1.52±0.48 | -0.03734 |
| 165 | GlcAβ1-3Galβ-sp3 | 1.45±0.40 | 0.137664 |
| 166 | GlcAβ1-6Galβ-sp3 | 1.61±0.36 | 0.09887 |
| 18J | 6-$H_2PO_3$Glc | 1.40+0.88 | 0.480741 |
| | **Glycosaminoglycans** | | |
| 12A | Neocarratetrose-41, 3-di-O-sulphate ($Na^+$) | 1.63±0.86 | 0.143975 |
| 12B | Neocarratetrose-41-O-sulphate ($Na^+$) | 1.69+0.81 | 0.217521 |
| 12E | Neocarraoctose-41, 3, 5, 7-tetra-O-sulphate ($Na^+$) | 1.67±0.91 | 0.102244 |
| 12G | ΔUA-2S-GlcNS-6S $Na_4$ (I-S) | 1.41±0.78 | 0.198534 |
| 12M | ΔUA-GlcNAc Na (IV-A) | 1.73+0.60 | 0.249342 |
| 13K | Chondroitin sulfate | 2.71±1.74 | 0.836093 |
| 13L | Dermatan sulfate | 2.70±0.58 | 0.852452 |
| 13N | HA—4 | 1.79±0.64 | 0.622733 |
| 14I | HA 1600000 Da | 2.67±0.94 | 0.846553 |

types recognised by both *M. ulcerans* cells and purified MUL_3720 to aid in the identification of the hierarchy of glycan recognition. The SPR analyses reconfirmed that both MUL_3720 and *M. ulcerans* cells interact with high affinity ($K_D$ in the low μM range) with sulfated glycans and GAGs (Table 2).

In the SPR analysis, MUL_3720 showed the strongest binding to the chondroitin 6S-disaccharide ΔUAα 1-3GalNAc-6S. However, due to lack of availability, the best MUL_3720 binding glycans from the array analysis (compounds 151, 161, 182, 189 and199) were not analysed. There is a large difference between the complex 6S-polymer and the 6S-disaccharide of chondroitin, with a greater than 10-fold higher affinity to the 6S-disaccharide for both the *M. ulcerans* cells and MUL_3720. The binding to α2–6 sialyated glycans on the glycan array were confirmed to be differential between *M. ulcerans* cells and MUL_3720 with the whole bacteria binding to the α2-6Neu5Ac terminating glycan (Neu5Acα2-6Galβ1-4GlcNAc) with ~4600-fold higher affinity than MUL3720 (Table 2).

## *M. ulcerans* cells and MUL_3720 show high affinity interactions with human keratin extract

Reports studying keratins isolated from primary human cells rather than cancer cell lines, have reported the presence of keratan sulfate glycans in human keratin extracts [26]. Due to this

**Table 2. SPR results for the binding of *M. ulcerans* cells and MUL_3720 protein to selected glycans.** Shown are dissociation equilibrium constants ($K_D$) of the interactions between free oligosaccharides and captured *M. ulcerans* cells and MUL_3720 protein; NCDI: no concentration-dependent interaction detected up to a maximum concentration of 20 μM. See S2 Data for representative sensorgrams.

| Oligosaccharide/glycoprotein | *M. ulcerans* | MUL_3720 |
|---|---|---|
| Fucα1-2Gal | 1.10 μM ± 0.23 | NCDI |
| Fucα1-2Galβ1-4(Fucα1–3)GlcNAc | 4.47 μM ± 0.62 | NCDI |
| chondroitin 6S (ave mol weight 14kDa) | 1.93 μM ± 0.44 | 4.68 μM ± 0.75 |
| chondroitin 6S-disaccharide ΔUAα 1-3GalNAc-6S | 171 nM ± 28 | 99.2 nM ± 35 |
| heparin (ave mol weight 14kDa) | 1.48 μM ± 0.19 | 1.29 μM ± 0.078 |
| Neu5Acα2-6Galβ1-4GlcNAc | 1.49 nM ± 0.36 | 6.90 μM ± 3.2 |
| Fucα1-2(3-O-Su)Galβ1-3(Fucα1–4)GlcNAc | 12.8 μM ± 3.7 | 17.4 μM ± 1.9 |
| Manα1-3Man | 49.8 nM ± 11 | NCDI |
| Manα1-6(Manα1–3)Manα1-6(Manα1–3)Man | 962 nM ± 90 | NCDI |
| Human keratin extract (ave mol weight 63kDa) | 37.9 nM ± 12 | 4.78 nM ± 1.1 |

observation, we analysed the interaction of keratin extracted from human epidermis (mixed keratins with average molecular weight of 64.1 kDa) with MUL_3720 and *M. ulcerans* cells. The cells bound to the keratin extract with a $K_D$ of 37.9 nM (Table 2), on par with interactions observed for the glycans Neu5Acα2-6Galβ1-4GlcNAc and Manα1-3Man. MUL_3720 showed a nearly 10 fold higher affinity to the keratin extract ($K_D$ 4.78 nM; Table 2). To determine whether the binding of MUL_3720 to keratin was via the sulfated N-linked glycans, we enzymatically removed N-linked glycosylation from the human keratin extract. After PNGase F treatment of the keratin extract we compared the staining using silver staining of glycans (S2 Fig lanes 3 and 4) and protein (S2 Fig lanes 6 and 7); this analysis shows that the keratin extract comprises a heterogeneous mixture of this glycoprotein, ranging from 26-43kDa. Treatment with the PNGase F removed all N-glycans leaving a single main band of reduced molecular mass. To confirm that the glycosylated portion of the keratin extract was the target for MUL_3720 we conducted competition studies between MUL_3720, ΔUAα 1-3GalNAc-6S and keratin in SPR studies (Fig 1). These data indicate strongly for an overlap of the binding sites for the 6S-disaccharide and the keratin extract, as the data for the binding to the combined ligands was closer to a combined average than to an additive signal or to one outcompeting the other.

## Discussion

We observed that *M. ulcerans* cells bind to a far smaller group of glycans present on our glycan array than other bacteria that have been assayed previously, including *Campylobacter jejuni* and *Neisseria meningitidis* [19,20]. The glycan-binding profile of *M. ulcerans* demonstrated that this skin pathogen recognises a range of negatively charged glycans common to sulfated GAGs, sialoglycoconjugates and keratans. Much of the binding is to negatively charged modifications (sulfation/sialylation) of carbon-6 of Gal or GalNAc. The whole *M. ulcerans* cells were found in SPR analyses to have the highest affinity among the tested glycans for the sialylated Neu5Acα2-6Galβ1-4GlcNAc (Table 2). While this exact glycan was not bound on the array, smaller mono/disaccharide versions of it were tested on the array. Testing using the larger free trisaccharide can reflect the results of smaller sugars. This is due to the flexibility of a glycan that has not been anchored on the non-reducing end, the end normally attached to a protein/lipid/array surface. The non-reducing GlcNAc of this sugar is not in any fixed anomeric configuration (α/β) due to it not being linked. All the glycans on the array have a fixed anomeric configuration and orientation dictated by the linkage to the spacer. This spacer effect

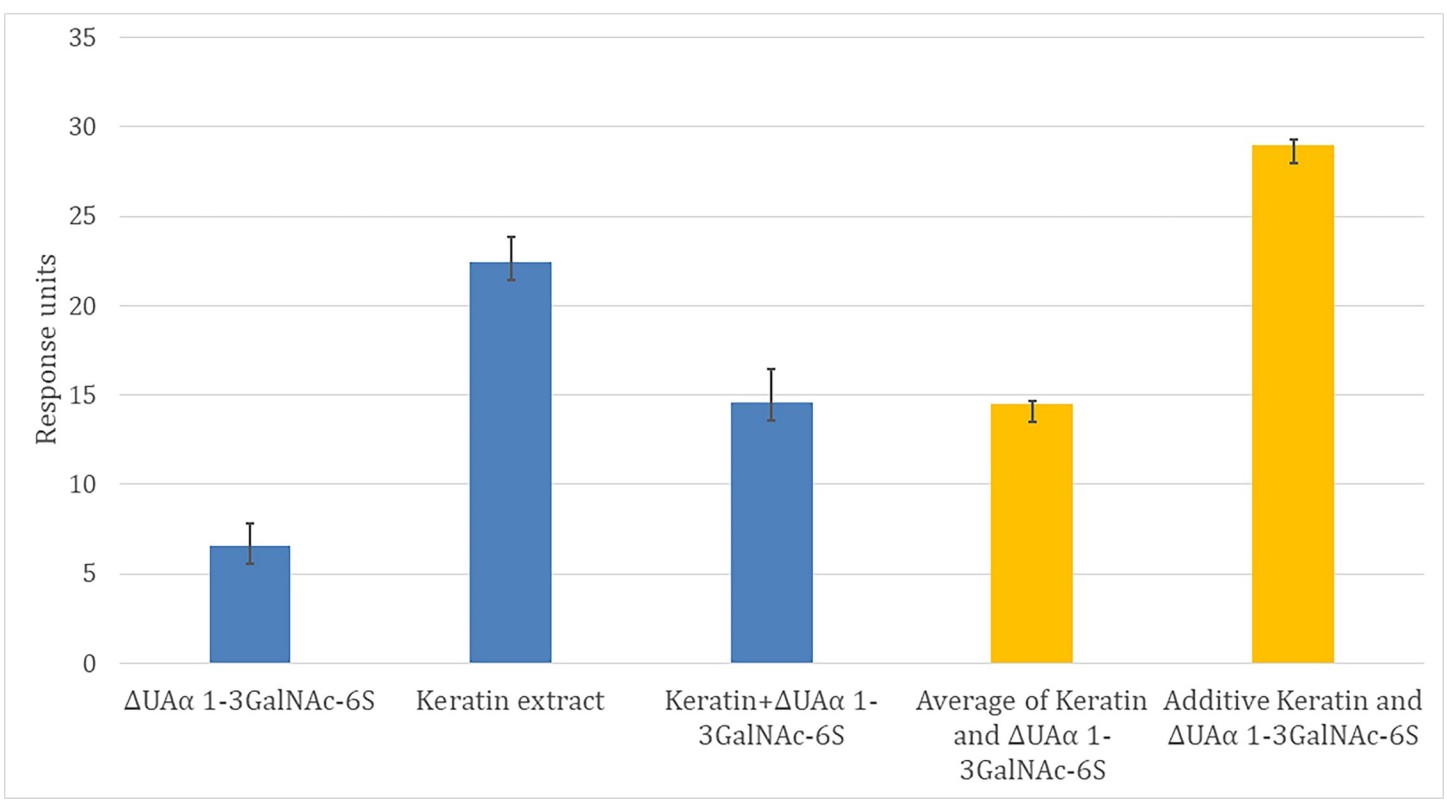

**Fig 1. SPR competition analysis of ΔUAα 1-3GalNAc-6S and keratin extract.** Values are the average maximum endpoint response units obtained individually and in competition at 10x $K_D$. The two columns in orange represent the expected values for either a shared binding site (average signal for keratin and ΔUAα 1-3GalNAc-6S) or separate binding sites (additive signal for keratin and ΔUAα 1-3GalNAc-6S).

has been observed to effect the binding of molecules to glycans printed on arrays previously [27,28]. It is due to the effect of presentation that testing with a second method such as SPR is so important. In human skin Neu5Acα2–6 is known to be associated with eccrine sweat glands [29], which may be a potential invasion site for *M. ulcerans* [30]. The high affinity binding of this structure was not due to MUL_3720, and is likely to be mediated by another *M. ulcerans* lectin.

Furthermore, the bacilli also bind to mannose structures, which are a major component of *N*-linked glycans [31]. Mannose glycans play an important role in the skin, with mannose-containing *N*-linked glycans important for skin homeostasis cell-cell adhesion and cell motility [32,33]. The surface expression of mannose-containing *N*-linked glycans on cells of the skin may provide an initial attachment site for *M. ulcerans*, but the bacterial surface factors involved in this binding remain to be determined.

Mycobacterial species produce numerous cell surface glycan structures with a large proportion of their surface glycosylation being lipid-linked sugars. Recently, glycan-glycan interactions have been identified as a mechanism for pathogen association with host cells [9,20,34–37]. Surface glycans such as the lipo-oligo/polysaccharide of Gram-negative bacteria have been shown to bind to host surface glycans [9,34,35,38,39]. The extent and diversity of the glycolipids produced by mycobacterial species may thus provide another adherence factor that can explain the glycan recognition of *M. ulcerans* cells in the glycan array experiment.

The *M. ulcerans* protein MUL_3720 recognises a subset of the determined *M. ulcerans* glycan binding profile, including sulfated galactose and sulfated GalNAc structures. These

sulfated glycans are primarily found as components of the GAGs chondroitin (repeating Gal-NAc(±2,4,6S)β1-3GlcA), dermatin (repeating GalNAc(±2,4,6S)β1-3IdoA) and as keratan sulfate glycosylations. Keratan sulfate glycans come in a range of forms including GalNAcα(±3,6S)1-3Gal(NAc(±6S)β [25], Gal(±6S)β1-3GlcNAc(±6S)/GalNAc(±6S) [24,26], and keratan sulfate glycans containing Gal(±6S)β1-3GlcNAc(±6S)/GalNAc(±6S), present on proteins including lumican, keratocan, and mimecan. The MUL_3720 protein was predicted to be a mannose binding lectin [10], however, the glycan array analysis revealed that MUL_3720 is binding to sulphated GAGs and not to mannose structures. MUL_3720 is an orthologue of a predicted lectin found in multiple other mycobacterial species including *M. marinum*, *M. basiliense*, *M. riyadhense*, *M. attenuatum*, *M. gastri*, *M. pseudokansasii*, *M. innocens* and *M. persicum*, with the *M. marinum* orthologue being most similar (99% sequence identity [10]) to the *M. ulcerans* protein (S1 Fig). Like *M. ulcerans*, *M. marinum* infections in humans are also typically limited to the skin [40–43], indicating the possibility that *M. marinum* MMAR_3773 is targeting similar glycans. *M. marinum* also causes a tuberculosis like infection in various fish species [40] and sulphated GalNAc structures such as those found in keratan and chondroitin sulfate are the most common sulfated GAGs in many fish species [44–46]. The data on MUL_3720 may be indicative for a role of MMAR_3773 in *M. marinum* infections in both humans and fish.

*M. ulcerans* infects skin, one of the keratin/keratan-rich tissues of the body, with the epidermal layer being made of keratinocytes and the dermal layer having high keratan sulfate areas around hair follicles and sweat glands/ducts [47,48]. Furthermore, human skin keratin extracts have indicated that keratin is either strongly associated with or is decorated by keratan sulfate glycosylation [26] (S2 Fig). Keratan sulfate proteoglycans such as lumican are crucial components of the dermal layer of skin [49,50]. Mice lacking lunican have skin laxity and fragility caused by improper organisation of collagen fibrils [49]. Keratin and keratan sulfate-containing proteins have a wide range of functions throughout the body, including intimate involvement in wound healing [24,47,50–52]. MUL_3720 binds with high affinity ($K_D = 4.78$ nM) to keratin and this binding may have a role in adherence, tissue tropism, and the formation of extracellular clusters of the pathogen in the skin. Clustering of the bacteria appears to be a prerequisite for the formation of a protective cloud of mycolactone that prevents elimination of the bacteria by phagocytes [8]. Most BU patients have only a single lesion and formation of satellite lesions is rare. This is indicative for retention of the bacteria to the infected skin area and the abundant protein MUL_3720 may play an important role in tissue tropism and sequestration.

## Supporting information

**S1 Data. Red indicates binding.** Binding is determined by positive interaction in three replicate array experiments. Positive interactions are determined by a background subtracted fluorescence value significantly above background subtracted fluorescence of negative control spots (average background fluorescence from 20 spots + 3 standard deviations).
(PDF)

**S2 Data. Representative sensorgrams of SPR analysis.**
(PDF)

**S1 Table. Supplementary glycan microarray document based on MIRAGE guidelines DOI: 10.1093/glycob/cww118.**
(DOCX)

**S1 Fig. Evolutionary analysis of mycobacterial MUL_3720 orthologues.**
(TIF)

**S2 Fig. SDS-PAGE analysis of PNGase F treatment of keratin extract.** Lane 1 DNA ladder, Lane 2 and 5 NEB prestained blue protein ladder. Lane 3 and 6 Keratin extract without PNGaseF treatment. Lane 4 and 7 Keratin extract with PNGase F treatment. PNGase is labelled on the right side of the gel image.
(TIF)

## Author Contributions

**Conceptualization:** Christopher J. Day, Katharina Röltgen, Gerd Pluschke, Michael P. Jennings.

**Data curation:** Christopher J. Day.

**Formal analysis:** Christopher J. Day, Gerd Pluschke.

**Funding acquisition:** Gerd Pluschke, Michael P. Jennings.

**Investigation:** Christopher J. Day.

**Methodology:** Christopher J. Day, Katharina Röltgen, Gerd Pluschke, Michael P. Jennings.

**Resources:** Katharina Röltgen.

**Supervision:** Michael P. Jennings.

**Writing – original draft:** Christopher J. Day, Gerd Pluschke, Michael P. Jennings.

**Writing – review & editing:** Christopher J. Day, Katharina Röltgen, Gerd Pluschke, Michael P. Jennings.

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
