## [Decision Letter · Decision Letter 0]

5 Nov 2019

Dear Prof. Jennings:

Thank you very much for submitting your manuscript "The cell surface protein MUL_3720 confers binding of the skin pathogen Mycobacterium ulcerans to sulfated glycans found on keratin." (#PNTD-D-19-01510) for review by PLOS Neglected Tropical Diseases. Your manuscript was fully evaluated at the editorial level and by independent peer reviewers. The reviewers appreciated the attention to an important problem, but raised some substantial concerns about the manuscript as it currently stands. These issues must be addressed before we would be willing to consider a revised version of your study. We cannot, of course, promise publication at that time.

We therefore ask you to modify the manuscript according to the review recommendations before we can consider your manuscript for acceptance. Your revisions should address the specific points made by each reviewer. 

When you are ready to resubmit, please be prepared to upload the following:

(1) A letter containing a detailed list of your responses to the review comments and a description of the changes you have made in the manuscript.

(2) Two versions of the manuscript: one with either highlights or tracked changes denoting where the text has been changed (uploaded as a "Revised Article with Changes Highlighted" file); the other a clean version (uploaded as the article file).

(3) If available, a striking still image (a new image if one is available or an existing one from within your manuscript). If your manuscript is accepted for publication, this image may be featured on our website. Images should ideally be high resolution, eye-catching, single panel images; where one is available, please use 'add file' at the time of resubmission and select 'striking image' as the file type. 

Please provide a short caption, including credits, uploaded as a separate "Other" file. If your image is from someone other than yourself, please ensure that the artist has read and agreed to the terms and conditions of the Creative Commons Attribution License at http://journals.plos.org/plosntds/s/content-license (NOTE: we cannot publish copyrighted images). 

(4) If applicable, we encourage you to add a list of accession numbers/ID numbers for genes and proteins mentioned in the text (these should be listed as a paragraph at the end of the manuscript). You can supply accession numbers for any database, so long as the database is publicly accessible and stable. Examples include LocusLink and SwissProt.

(5) To enhance the reproducibility of your results, we recommend that you deposit your laboratory protocols in protocols.io, where a protocol can be assigned its own identifier (DOI) such that it can be cited independently in the future. For instructions see http://journals.plos.org/plosntds/s/submission-guidelines#loc-methods

While revising your submission, please upload your figure files to the Preflight Analysis and Conversion Engine (PACE) digital diagnostic tool, https://pacev2.apexcovantage.com/ PACE helps ensure that figures meet PLOS requirements. To use PACE, you must first register as a user. Then, login and navigate to the UPLOAD tab, where you will find detailed instructions on how to use the tool. If you encounter any issues or have any questions when using PACE, please email us at figures@plos.org.

We hope to receive your revised manuscript by Jan 04 2020 11:59PM. If you anticipate any delay in its return, we ask that you let us know the expected resubmission date by replying to this email.

To submit a revision, go to https://www.editorialmanager.com/pntd/ and log in as an Author. You will see a menu item call Submission Needing Revision. You will find your submission record there. 

Sincerely,

Bradley R. Borlee

Guest Editor

Richard Phillips

Deputy Editor

Then Manuscript by Day et. al presents studies that contribute to a greater understanding of glycan binding by Mycobacterium ulcerans to better understand cellular attachment to host targets. All of the reviewers had comments and suggestions that would improve a revised version of the manuscript and enhance the impact of the studies presented. It is essential that the quantitative data associated with Table 1 should be included in the main body of the manuscript in a format that allows the reader to make quantitative comparisons. The numbers could be added to the table in addition to actual values for the background spots to assess whether the cutoff is appropriate. Additional discussion of the bioinformatics associated with MUL3720 that includes homology to other mycobacterial proteins would be beneficial to bring the reader up to date concerning what is known about this protein beyond the authors’ previous publication. Additionally the sensograms produced by the Biacore SPR instrument should be made available as part of this submission for assessment of the kinetics of dissociation as this provides a measure of avidity which may be equally important for the biological activity of MUL3720 binding. One reviewer was particularly concerned that the title was inaccurate in the conclusions reached and that “Mycobacerium” was misspelled, so it would be essential to alleviate these concerns.

Reviewer's Responses to Questions

**Key Review Criteria Required for Acceptance?**

**Methods**

-Are the objectives of the study clearly articulated with a clear testable hypothesis stated?

-Is the study design appropriate to address the stated objectives?

-Is the population clearly described and appropriate for the hypothesis being tested?

-Is the sample size sufficient to ensure adequate power to address the hypothesis being tested?

-Were correct statistical analysis used to support conclusions?

-Are there concerns about ethical or regulatory requirements being met?

Reviewer #1: There are some significant issues with the presentation. The protein expression protocol is a verbatim repetition of an identical paragraph in reference 10. The actual synthetic sequence and the primers for PCR are not given either there or here and documentation of the protein purification not shown.

Reviewer #2: (No Response)

Reviewer #3: The authors describe a nice study designed to identify the major glycans interacting with Mycobacterium ulcerans and then test the hypothesis that, based on sequence motifs, MUL_3720 participates in glycan binding. The authors methods for initial identification are straight forward. The authors provide enough information and data for clear understanding of the methods. The manuscript would benefit from an additional method where binding is competed; one, due to the conjecture of multiplicity of ligands interacting with the 64 glycans identified for whole cells, and more importantly, to better determine the specific glycans interacting with MUL_3720, and whether there is one, or multiple interaction sites on the protein. This is important because the glycan motif cannot be discerned from the glycans identified alone (there is no consensus motif).

**Results**

-Does the analysis presented match the analysis plan?

-Are the results clearly and completely presented?

-Are the figures (Tables, Images) of sufficient quality for clarity?

Reviewer #1: Please note comments below regarding presentation of the glycan array results.

Reviewer #2: The quantitative data associated with Table 1 should be included in the main manuscript 

See additional comments in summary

Reviewer #3: The authors present data demonstrating that whole M. ulcerans binds to 64 glycans, all of which resemble sulfated glycoconjugates found on epidermal and dermal cells. The authors further present data demonstrating that 7 of these 64 glycans bound to MUL_3720. The data may be a little misleading for a number of these 7 glycans. This is because ~ 1 X 10E6 cells were used versus 2 microgram of recombinant protein. 10E6 mycobacterial cells would equate to roughly 3 micrograms of dehydrated cell biomass, and then at best, 2 micrograms of total protein. Even if MUL_3720 is 10% of the total protein, that would be 0.2 microgram of the total. Thus where the glycan arrays show similar binding and disassociation constants, these are likely not bona fide binders for MUL_3720 (the specific activity would be higher relative to whole cells). The binding comparison to keratin, however, is quite nice and supports their hypothesis.

**Conclusions**

-Are the conclusions supported by the data presented?

-Are the limitations of analysis clearly described?

-Do the authors discuss how these data can be helpful to advance our understanding of the topic under study?

-Is public health relevance addressed?

Reviewer #1: (1) There are significant concerns about the conclusion that the primary target for the mycobacterial lectin is glycosylated keratin. It is indicated that binding it to keratan sulfate oligosaccharides attached to keratin, with reference 15 cited as the basis for claiming the keratin protein is conjugated with glycosaminoglycans. Beyond the work in this 1993 manuscript, there appears to be no other published information on this suggestion and a recent comprehensive review of post-translational modifications of keratins, Nature Reviews Molecular Cell Biology 15, 163–177 (2014), does not mention keratan sulfate. The conclusion in reference 15 is based on immunological analysis, with no biochemical follow-up, and the conclusions is nuanced: "These results demonstrate that a portion of the cytoplasmic anti-keratan sulfate immunoreactivity is due to keratins that are glycosylated with carbohydrates that contain keratan sulfate epitopes or that keratan sulfate-containing molecules bind or co-migrate in SDS-polyacrylamide gels with cytokeratins." So even at the time the work was published, the possibility of contamination was highlighted. The lack of any further evidence for such a modification of keratin, as well the difficulty of explaining how a Golgi-specific modification could occur on a cytoplasmic protein, bears out the original caution about the interpretation of these results. The material used in the assays was a commercial preparation of mixed keratins that does not seem to have been further characterized to eliminate the very real possibility of contamination with one or more of the several proteoglycans that bear keratan sulfate chains. It is also not clear how it was kept in solution in the surface plasmon resonance assay, since 8 M urea is required to keep keratins soluble. Given the multiple uncertainties associated with these experimental results and their interpretation, it is premature to conclude that the target for MUL_3720 has been identified.

(2) The presentation of the glycan array results in the main manuscript is without any quantitative information. Examination of the complete results in supplemental information indicates that the dynamic range, from lowest to highest signals, is very limited compared other experiments with this array, such as in reference 16. The actual values are not given for the background spots, making it further difficult to assess whether the cutoff is appropriate.

(3) There is no comment on a key inconsistency between the array results and the binding competition: extremely high (nM) affinity binding of NeuAc2-6-N-acetyllactosamine to the fixed bacteria, and this ligand has one of the highest (microM) affinities for the expressed protein as reported in Table 2, while this identical oligosaccharide appears as glycan 300 on the array but shows only low background binding in each case. The absence of congruence between two experiments is a concern, particularly since as it stands the results in Table 2 suggest that the common feature of high affinity ligands for MUL_3720 is a negative charge: all the tested ligands with negative charge compete, those without charge do not compete. Additional ligands would need to be tested in this format to make a convincing case for specific binding.

(4) In Table 2, the quantification of affinities based on molar values for polysaccharides such as heparin and chondroitin sulfate is confusing, since these polysaccharides do not have defined molecular weights. The presentation of only "positive" results in Table 1 also masks the fact that the full results in the supplemental table show that other glycans containing similar sulfated residues do not bind. No effort is made to pick apart the differences. The fact that binding is seen to all three different types of glycosaminoglycans, keratan sulfate, chondroitin sulfate and heparin is hard to reconcile with binding to specific sulfate disaccharides.

(5) It is suggested on page 8 that "glycolipids produced by mycobacterial species may provide another adherence factor that can explain the glycan recognition of M. ulcerans cells in the glycan array experiment" seem to posit binding of glycolipids in the mycobacteria to sugars on the array. Such an interaction would be without precedent.

Reviewer #2: The limitations of the analyses need to be addressed

Reviewer #3: Again, this is a nice paper. It provides quite a nice insight regarding M. ulcerans pathogenesis/infection establishment. The methods need to have some improved rigor, and the conclusions regarding the results need to be provided in the context of the limitations of the comparisons from the glycan arrays. The conclusions should include the need to perform additional glycan studies to identify the bona fide interaction between keratin and the protein.

**Editorial and Data Presentation Modifications?**

Reviewer #1: The reference list is also in poor condition, with the names of authors missing in reference 2 and an incomplete citation in reference 19 as well as inconsistent formatting.

Reviewer #2: (No Response)

Reviewer #3: I think the data are nicely presented.

**Summary and General Comments**

Reviewer #1: (No Response)

Reviewer #2: The manuscript provides evidence that the M. ulcerans protein MUL3720 binds with keratin, sulfated glycans and glycosaminoglycans. The analyses performed are relatively straight forward and included the binding of M. ulcerans and MUL3720 to glycan arrays, and MUL3720 interactions with a variety of purified glycoproteins and glycans by surface plasmon resonance (SPR). The studies do not, however, demonstrate 1) that MUL3720 is the primary keratin binding ligand of M. ulcerans or that this protein is a factor in the colonization of skin based on keratin binding. Additional concerns with the manuscript are directed at the rationale for the selection of targets and the data presented. 

1) The significance of the work presented if the authors could demonstrate that purified MUL3720 is able to prevent the binding of M. ulcerans to keratin.

2) The rationale for targeting MUL3720 for glycan binding over other M. ulcerans proteins should be articulated. Is this the only protein of this bacterium with a predicted lectin domain?

3) A deeper discussion of the bioinformatics associated with MUL3720 that includes homology to other mycobacterial proteins would be beneficial to the reader. 

4) The glycan Neu5Ac�2-6Gal�1-4GlcNAc greater affinity than the sulfated glycan tested. The significance of the interaction with this glycosaminoglycan is not addressed by the authors. 

5) The Neu5Ac�2-6Gal�1-4GlcNAc used in SPR studies is not noted as a glycan tested by glycan array in Table 1. Thus, it is unclear whether this glycan was not included in the array or if it was not found to bind M ulcerans or MUL3720 using the array technology.

6) The selection of specific glycans to test for MUL3720 binding by SPR does not have a strong rationale. 

7) The discussion specifically mention sulfated glycan structures Array ID# 179, 161, and 151, that were found to bind to MUL3720 using the array technology. These glycans were not tested by SPR.

8) The sensograms produced by the Biacore SPR instrument allows for assessment of the kinetics of dissociation. This provides a measure of avidity which may be equally important for the biological activity of MUL3720 binding. This data should be provided.

9) The quantitative data for table 1 should be provided in the main manuscript.

Reviewer #3: The study was very straightforward, and just needs to be a little more comprehensive prior to publication. Thus the competitive binding assays were suggested; the results of which would provide important data elements regarding the glycan-protein interactions. Finally, since minimally 10 fold excess recombinant protein was used relative to the whole cell assays, the data should be presented in the context of this potential bias.

PLOS authors have the option to publish the peer review history of their article (what does this mean?). If published, this will include your full peer review and any attached files.

Reviewer #1: No

Reviewer #2: No

Reviewer #3: Yes: Karen M. Dobos

---

## [Decision Letter · Decision Letter 1]

13 May 2020

Dear Prof. Jennings,

Thank you very much for submitting your manuscript "The cell surface protein MUL_3720 confers binding of the skin pathogen Mycobacterium ulcerans to sulfated glycans and keratin." for consideration at PLOS Neglected Tropical Diseases. As with all papers reviewed by the journal, your manuscript was reviewed by members of the editorial board and by several independent reviewers. The reviewers appreciated the attention to an important topic. Based on the reviews, we are likely to accept this manuscript for publication, providing that you modify the manuscript according to the review recommendations. 

Two of three reviewers (#2 and #3) found the revised manuscript had addressed many of their concerns. These reviewers did point out a few minor issues that need to be addressed during revision. Reviewer No 1 also has some concerns. Please address the concerns with citations/references for reviewer #1.

Sincerely,

Bradley R. Borlee

Guest Editor

Richard Phillips

Deputy Editor

Two of three reviewers (#2 and #3) found the revised manuscript had addressed many of their concerns. These reviewers did point out a few minor issues that need to be addressed during revision. Reviewer No 1 also has some concerns. Please address the concerns with citations/references for reviewer #1.

Reviewer's Responses to Questions

**Key Review Criteria Required for Acceptance?**

**Methods**

-Are the objectives of the study clearly articulated with a clear testable hypothesis stated?

-Is the study design appropriate to address the stated objectives?

-Is the population clearly described and appropriate for the hypothesis being tested?

-Is the sample size sufficient to ensure adequate power to address the hypothesis being tested?

-Were correct statistical analysis used to support conclusions?

-Are there concerns about ethical or regulatory requirements being met?

Reviewer #1: No change to previous comment.

Reviewer #2: (No Response)

Reviewer #3: The authors have satisfactorily addressed all concerns related to the methods used in this study. The reviewer appreciates the inclusion of raw and supplementary data in support of the methods used.

**Results**

-Does the analysis presented match the analysis plan?

-Are the results clearly and completely presented?

-Are the figures (Tables, Images) of sufficient quality for clarity?

Reviewer #1: No change to previous comment.

Reviewer #2: (No Response)

Reviewer #3: The authors description of the results are accurate and acceptable; especially given the limitations of the current technology, annoted information for M. ulcerans and related species, and the scope of this study.

**Conclusions**

-Are the conclusions supported by the data presented?

-Are the limitations of analysis clearly described?

-Do the authors discuss how these data can be helpful to advance our understanding of the topic under study?

-Is public health relevance addressed?

Reviewer #1: No.

(1) Fundamentally, the response to the previous comments is disappointing. Rather than providing rigorous evidence about the purity of the keratins used in these studies, the title and abstract have been somewhat nuanced about the connection between keratins and keratan sulfate. As it stands, the description "glycans associated with human skin keratin" is vague and could be misleading, as it might just mean glycans that contaminate the keratin preparation.

(2) Several specific points raised in the first review have not been addressed:

Original Point 1

(a) Reference 26 has been added and cited as showing keratan sulfate is attached to keratins. This reference reports a novel form of glycosylation and specifically states "the glycans described here do not resemble keratan sulfate." Citing it in this way misrepresents the conclusion completely.

(b) The revision also does not address the point in the original review that it is also not clear how keratins were kept in solution in the surface plasmon resonance assay, given the need for 8 M urea keep them keratins.

Original Point 4

(a) In Table 2, the quantification of affinities based on molar values for polysaccharides such as heparin and chondroitin sulfate is confusing, since these polysaccharides do not have defined molecular weights. There is no response to this commentl.

(b) The fact that binding is seen to all three different types of glycosaminoglycans, keratan sulfate, chondroitin sulfate and heparin is hard to reconcile with binding to specific sulfate disaccharides. The response "The lack of binding to some 6-sulfated glycans on the array is consisted [sic] with our array and subsequent, more sensitive SPR studies" is vague. Lowering the cutoff to a level required to detect binding of many glycans containing sulfated galactose and GalNAc, so they are not "false negatives," would mean that roughly 90% of the glycans on the array would be treated as positive for binding.

Reviewer #2: (No Response)

Reviewer #3: The conclusions are much improved, and include a nice discussion of the limitations of this study and continued questions to be addressed in subsequent studies.

**Editorial and Data Presentation Modifications?**

Reviewer #1: The reference list still contains a mixture of journal names written out, often without capitalization, and abbreviations.

Reviewer #2: (No Response)

Reviewer #3: A few minor comments to note:

1. In the abstract, "SPR" is stated as an abbreviation. SPR should be spelled out in the abstract. 

2. In the methods, it was hard to find "Keratin isolated from human epidermis (mixed keratins...) from Sigma Adrich. The authors should confirm the description of what was ordered, or provide a catalog #.

3. In the results, in the legend for Table 1, presented is misspelled as "presende"

**Summary and General Comments**

Reviewer #1: The underlying issues raised in the previous review have not been addressed.

Reviewer #2: The authors have appropriately addressed my concerns and comments. It is noted that 1) the abbrevriation PNG in the third line of the of the introduction should not be replaced with the full name and 2) the authors sound be constant in the use of significant figures in Table 2.

Reviewer #3: The authors revised manuscript is considerably improved from the initial submission.

PLOS authors have the option to publish the peer review history of their article (what does this mean?). If published, this will include your full peer review and any attached files.

Reviewer #1: No

Reviewer #2: No

Reviewer #3: Yes: Karen M. Dobos
---

## [Editor Report · Decision Letter 2]

13 Jan 2021

Dear Prof. Jennings,

We are pleased to inform you that your manuscript 'The cell surface protein MUL_3720 confers binding of the skin pathogen Mycobacterium ulcerans to sulfated glycans and keratin.' has been provisionally accepted for publication in PLOS Neglected Tropical Diseases.

Best regards,

Bradley R. Borlee

Associate Editor

Richard Phillips

Deputy Editor

---

## [Editor Report · Acceptance letter]

10 Feb 2021

Dear Prof. Jennings,

We are delighted to inform you that your manuscript, "The cell surface protein MUL_3720 confers binding of the skin pathogen Mycobacterium ulcerans to sulfated glycans and keratin.," has been formally accepted for publication in PLOS Neglected Tropical Diseases.

Best regards,

Shaden Kamhawi

co-Editor-in-Chief

Paul Brindley

co-Editor-in-Chief
